# Potential Impact of Climate Change on Salmonid Smolt Ecology

**Teppo Vehanen** *[ID], **Tapio Sutela** [ID] and **Ari Huusko**

Natural Resources Institute Finland, Latokartanonkaari 9, 00790 Helsinki, Finland; tapio.sutela@luke.fi (T.S.); ari.huusko@luke.fi (A.H.)

\* Correspondence: teppo.vehanen@luke.fi

**Abstract:** The migratory life history of anadromous salmonids requires successful migration between nursery, feeding, and spawning habitats. Smolting is the major transformation anadromous salmonids undergo before migration to feeding areas. It prepares juvenile fish for downstream migration and their entry to seawater. We reviewed the effects of climate change on smolt ecology from the growth of juveniles in fresh water to early post-smolts in the sea to identify the potential effects of climate change on migratory salmonid populations during this period in their life history. The focus was especially on Atlantic salmon. The shift in suitable thermal conditions caused by climate change results in Atlantic salmon expanding their range northward, while at the southern edge of their distribution, populations struggle with high temperatures and occasional droughts. Climatic conditions, particularly warmer temperatures, affect growth during the freshwater river phase. Better growth in northern latitudes leads to earlier smolting. Thermal refuges, the areas of cooler water in the river, are important for salmonids impacted by climate change. Restoring and maintaining connectivity and a suitably diverse mosaic habitat in rivers are important for survival and growth throughout the range. The start of the smolt migration has shifted earlier as a response to rising water temperatures, which has led to concerns about a mismatch with optimal conditions for post-smolts in the sea, decreasing their survival. A wide smolt window allowing all migrating phenotypes from early to late migrants' safe access to the sea is important in changing environmental conditions. This is also true for regulated rivers, where flow regulation practices cause selection pressures on migrating salmonid phenotypes. The freshwater life history also affects marine survival, and better collaboration across life stages and habitats is necessary among researchers and managers to boost smolt production in rivers. Proactive measures are recommended against population declines, including sustainable land use in the catchment, maintaining a diverse mosaic of habitats for salmonids, restoring flow and connectivity, and conserving key habitats.

**Keywords:** climate change; salmonids; *Salmo*; rivers; fresh water; migration

**Key Contribution:** This paper makes a valuable contribution to understanding how climate change affects the key life stage of salmonids, smolting, and smolt migration. Smolting prepares entry from fresh water to salt water. Salmonids are a group of species with huge cultural and economic significance.





## 1. Introduction

Anadromous and potamodromous salmonids migrate from their natal river to a feeding environment before returning for reproduction [1–3]. Migration enables fish to exploit many temporally productive and spatially discrete habitats with various fitness benefits (e.g., growth, reproduction, predator avoidance) [4]. Migratory life history requires unrestricted migration routes between nursery, feeding, and spawning habitats [5]. During each life stage, salmonids utilize the habitat that is advantageous for them. Migration between habitats thus clearly has an adaptive value [6]. Nursery and feeding habitats differ in environmental characteristics, and migrations precede adaptive physiological

transformations and changes in the phenotype and behavior to be better suited for the new environment.

Smolting is the major transformation anadromous salmonids undergo before migration to feeding areas. Smolting prepares fish for downstream migration and entry to seawater. Atlantic salmon *Salmo salar* L., 1758 juveniles can stay in their natal river habitat to grow for 1–8 years before migrating [7,8]. Of the Pacific salmon, pink salmon *Oncorhynchus gorbuscha* (Walbaum, 1792) and chum salmon *Oncorhynchus keta* (Walbaum, 1792) can move almost directly after the emergence at the fry stage into seawater, while the others (masu salmon *Oncorhynchus masou* (Brevoort, 1856), *O. tshawytscha* (Walbaum, 1792), *O. nerka* (Walbaum, 1792), and steelhead (rainbow) trout *Oncorhynchus mykiss* (Walbaum, 1792)) spend one or more years in fresh water before migrating to the sea for feeding [7]. When smolting, the phenotype of fish changes as follows: the coloration of smolts becomes silvery, and the body shape becomes more streamlined [2]. This, with darkened fins, a dark back, and a white abdomen, camouflage the fish in the pelagic environment [5]. Behavioral changes include a loss of rheotaxis, and juveniles become more pelagic. Their tendency to group also increases [9]. Several physiological changes occur, for example, increased salinity tolerance, increased metabolism, and olfactory imprinting helping the fish locate their home stream on their return migration [10]. Environmental cues, e.g., photoperiod, temperature, and waterflow, regulate physiological changes and initiate migration [2,5]. Lake-living Atlantic salmon appear to smolts similarly to anadromous conspecifics ([11], but see [12]). This is apparently an inherited trait [13], even for salmon spending their entire life in fresh water, such as the Atlantic salmon residing in Lake Vänern, Sweden.

Both Atlantic and Pacific salmon populations have been in decline throughout their habitat ranges [13–15]. To reverse this trend, it is important to understand the role of different environmental and anthropogenic factors in the decline [16]. Numerous factors may impact population abundances negatively, and with the complex life history of migrating salmonids, the reasons are obviously multiple and difficult to unravel [17,18], although the ongoing climate warming appears particularly important, especially for low latitude populations. Anthropogenic activities have a long history of altering salmonid populations and, thus, smolt development and smolt migration. Smolts are sensitive to external impacts and behavior and survival during migrations [2]. Several anthropogenic activities may affect smolt development, behavior, and survival during migration, such as hydropower developments, land use, pollution, fish farming, and parasites like sea lice *Lepeophtheirus salmonis* (Krøyer, 1837) [2,5]. Temperature and flow interact with the other anthropogenic pressures to affect smolting and smolt migration.

Salmonids are a cold-water species. Global warming will generally have a major impact on their success. Historically, climatic variability has affected the patterns of abundance in Atlantic salmon and Pacific salmon populations [19–22]. Although estuarine and marine mortalities have been found to be important determinants of survival, marine mortality depends on factors acting in fresh water and during smolt migration [23]. Thorstad et al. [24] argue that the best strategy to mitigate the changing environmental conditions should be to ensure that the greatest number of wild smolts in the best condition migrate from rivers and coastal areas to feeding areas because mortality at sea is found to be density independent [25]. Survival at sea depends on the size of the smolts and environmental conditions when the smolts begin their sea sojourn. [18,23,26]. In research, it is important to address the links between river habitat conditions and the physiological requirements of salmonids during their juvenile life stages in freshwater habitats [27]. Climate change will continue to affect not only smolting and migration but also instream habitats across all seasons [27].

In this paper, we review climate change effects on (1) in-river habitat conditions in preparation for smolting, (2) the smolting process, (3) smolt migration, and (4) early post-smolt survival. Our focus is on *Salmo* spp., but when relevant, we also refer to the fish species in the Pacific salmon genus (*Onchorhynchus* spp.).

## 2. Climate Change and Salmonid Distribution

Human activities are estimated to have induced approximately 1.0 °C of global warming above pre-industrial levels (between 1880 and 2017), with a likely range of 0.8 °C to 1.2 °C. Global warming is likely to reach 1.5 °C in about 2030 if temperatures continue to increase at the current rate [28]. For example, a higher winter discharge, earlier snowmelt, and earlier onset of summer low flow periods are predicted throughout the range of Atlantic salmon [29,30].

Increasing global surface temperatures are very likely to lead to changes in precipitation and atmospheric moisture because of changes in atmospheric circulation, a more active hydrological cycle, and increases in the water-holding capacity throughout the atmosphere. Overall, global land precipitation has increased by about 2% since the beginning of the 20th century. There have been marked increases in precipitation in the latter part of the 20th century over northern Europe, though with a general decrease southward to the Mediterranean. Dry wintertime conditions over southern Europe and the Mediterranean and wetter-than-normal conditions over many parts of northern Europe and Scandinavia [31] are linked to the strong positive values of the North Atlantic Oscillation (NAO), with more anticyclonic conditions over southern Europe and stronger westerly winds over northern Europe (Ref. [32] conducted a review).

Northern Eurasia (north of approximately 40 °N) showed widespread and statistically significant increases in winter precipitation between 1921 and 2015, with values exceeding 1.2–1.6 mm mo$^{-1}$ per decade west of the Ural Mountains and along the east coast, while southern Europe exhibits coherent yet weaker amplitude drying trends that attain statistical significance over the eastern Mediterranean. These precipitation trends occur in the context of changes in the large-scale atmospheric circulation, with negative SLP (Sea Level Pressure) trends over northern Eurasia and positive SLP trends over the central North Atlantic extending into southwestern Europe [33].

The magnitude of climate change is considered to depend on the atmospheric load of the two most important greenhouse gases: carbon dioxide ($CO_2$) and methane ($CH_4$). The terrestrial biosphere plays an important role in the global carbon balance. In boreal zones, forests and peatlands are an essential part of the global carbon cycle. Recent temperature increases have been associated with increasing forest fire activity in Canada since about 1970 and exceptionally warm summer conditions in Russia during the 2010 fire season reviewed by [34].

Atlantic salmon is distributed from northern Portugal (42 °N) to the River Kara in northern Russia in Europe [35], and West Atlantic salmon is distributed from the Connecticut River to the Ungava region of northern Quebec. Southern Atlantic salmon populations have declined dramatically and face the highest risk of extinction as global warming moves its thermal niche northward [36]. The suitable thermal habitat for salmon is expected to extend northward with the invasion of new spawning, nursery, and feeding areas north of the species' present distributional range but with the loss of the most southern populations [37–40]. Indeed, salmon are already responding to warmer temperatures by expanding their range northward into the Arctic Ocean [41,42] and disappearing from the southern edge of their distribution area [7,40,43–45]. The population complex of Atlantic salmon in Europe has experienced a multidecadal decline in recruitment, resulting in the lowest population abundances observed since 1970 [46]. Atlantic salmon abundance and productivity show similar patterns of decline across six widespread regions of North America [47]. Abundance declined in the late 1980s and early 1990s, after which it remained stable at low levels. Climate-driven environmental factors such as changes in plankton communities and prey availability at warmer ocean temperatures were linked to the low productivity of North Atlantic salmon populations [47]. Landlocked European populations of salmonids are found in Norway, Sweden, Finland, and Russian Karelia [48–51]. The landlocked populations of salmon have declined throughout their distribution range [51,52]. Brown trout (*Salmo trutta* L. 1758) is native to Europe and Asia, where anadromous populations are found from Portugal to the White Sea [7]. It must be noted that the taxonomic

status of the brown trout species complex is challenging, and the high morphological and ecological diversity has led to the morphological description of populations belonging to species other than *S. trutta* in Europe [53]. In the future, the living conditions for trout will probably deteriorate in the southern part of the current distribution. In the northern part of their current distribution, global warming may improve feeding opportunities, growth, and survival conditions [7]. According to Filipe et al. [54], future brown trout distribution will become progressively and dramatically reduced in European watercourses. Their forecasts indicate that the greatest losses in suitable habitats will take place in southern Europe.

## 3. In-River Habitat Conditions in Preparation for Smolting

The most important climate-change-driven habitat changes that influence salmonid juveniles in rivers are changes in thermal and hydrological regimes [55–57]. These changes will affect how juveniles use their physical habitat and affect growth and survival.

Water temperature has various effects on the biology of salmonids. Thermal optima allow salmon to maximize growth; temperatures above thermal optima can stress fish and ultimately lead to mortality [58,59]. On a larger scale, northern populations are predicted to do better than southern populations under global warming [38,60,61], but even in the same river, the effects on different populations can vary [59]. Some northern populations can have an increase in parr recruitment and smolt production [61]. However, some Arctic salmonids are also already experiencing warm (>21 °C), physiologically challenging, migratory river conditions [62], and an increase in river water temperatures has already been observed in several rivers [63–65]. In general, high-latitude ecosystems are facing rapid warming, and cold-water fish will eventually be displaced by fish adapted to warmer water [66]. The range of temperatures at which fish survive or grow differs between development stages and salmonid species (for a review, see [7]). Atlantic salmon eggs survive between 0 and 16 °C, and alevins can develop normally up to 22 °C [67]. Growth takes place in temperatures between 6 °C and 22.5 °C, with maximum growth at around 16 °C, and the upper lethal temperature is 29.5 °C for parr but depends on the acclimatization temperature and the length of the acclimatization period. With the warming of surface waters, the risk of local extinctions will increase [68,69].

Smolt's characteristics are influenced by their earlier life in fresh water [23,70]. For example, the incubation of eggs at higher temperatures has resulted in fry with a reduced swimming performance or later returning adults [71,72]. According to Thompson and Beauchamp [73], the survival of steelhead trout in the marine environment can be driven by an overall higher growth rate established early in life in fresh water, which results in a larger size at smolt migration. Climate-induced instream thermal conditions affect parr size and the age of departure from the river [74]. For salmonid populations facing increased water temperatures, thermal heterogeneity in the river plays an important role in survival and growth [56,75]. The density of juveniles in thermal refuges has been found to increase after high-temperature effects [56]. Maintaining and restoring a diverse mixture of habitats and thermal refugia is important for salmonids impacted by climate change [76]. Thermal topology can also influence fish growth. Fish in the least complex network grew faster and were ready to smolt earlier than fish in the more spatially complex temperature network, i.e., in a river environment where the thermal diversity was higher [77]. Climate-induced high water temperatures can also interact with parr density, while in chinook salmon at low parr density, the effect of temperature on growth was positive, and at high densities, the relationship proved to be negative [75].

Especially in the southern margins of the salmonid distribution ranges, the availability of suitable cold-water environments becomes more important as the temperatures rise [78]. The temporal variability of these cold refuges is high; the most stable ones are typically groundwater seeps and cold-water tributaries [79]. For cold-water species like salmonids, headwater streams may become more important structural and thermal refuges. Headwaters are often less impacted by humans than the main streams (T. Vehanen, personal communication). On the other hand, high-elevation streams, especially those above snow-

lines, can be especially vulnerable to climate change because they are likely to experience the greatest snow–rain transition [80]. Stream size is a limiting factor for some salmonid species, but for species like coho salmon, differently sized streams can provide an alternative rearing habitat [81]. For brown trout, small streams are important spawning and nursery habitats [82,83]. Brown trout are well adapted and influenced by habitat variables associated with the size of small streams, especially with flow variations [82,84], and the population traits of anadromous brown trout from a small stream differ from those in larger rivers [72].

Seasonal flow is another key element impacted by climate change, contributing to the habitat quality of salmonid juveniles [85]. Climate change has already altered the hydrological regimes of rivers. The changes for Atlantic salmon and brown trout include frequent periods with extreme weather, i.e., low- and high-flow events, precipitation falling as rain and less as snow, and a decrease in the ice-covered period [7,86]. These changes can have a negative impact on freshwater salmon's instream habitat [80,87]. Extremes in waterflow can decrease recruitment and survival. Generally, the early life stages, i.e., the eggs, emerging alevins, fry, and young juveniles, experience the highest mortalities [88,89]. High-flow events during the emergence of fry from the gravel can cause the flushing of fry to unsuitable habitats. The preferences for physical habitat parameters like water velocity and depth vary seasonally [90]. Climate change-induced high or low flows cause variation in this habitat suitable for salmonid parr. Low flow conditions are also often associated with an extended duration of high water temperatures [87]. The minimum levels of river flow have been found to be regulators for parr survival and, hence, for smolt production in Atlantic salmon and brown trout [61,84]. It is also predicted that stream hydrology will change when winters get warmer, and increased fluctuations in winter discharge and temperatures may lead to repeated ice formation and breakup [91–93]. As winters get warmer, there is less snow, more rain, and higher winter discharges. These changes can negatively affect the growth and survival of juvenile salmonids during the winter [94]. The ability of the young salmonids to swim against strong currents is poor at low temperatures [7,95], and salmonid parr prefers relatively slow flow rates in the winter [96]. Increased rain on snow with a high flow can lead to the ice scouring of the streambed, which results in higher egg mortality [97]. The mortality of salmonid eggs and fry may become higher with climate change in northern rivers.

Water temperature and flow variation, the two important aspects of climate change, are interacting with anthropogenic activities, such as land use in the catchment to affect the fish community in rivers [98]. Anthropogenic activities have long altered migratory fish by closing pathways and creating challenging migration conditions for smolt. Climate change can further strengthen these human-caused effects. Climate change will intensify precipitation and flood events in all climate regions [99], but the difference at the regional scale can be high [100]. Increased precipitation intensity enhances suspended solid and nutrient loadings in rivers, especially in human-altered catchments [101]. Increased rainfall with land use (i.e., forestry, agriculture) will intensify the brownification of surface waters due to the increased loading of dissolved organic carbon from the catchments [102,103]. This widespread phenomenon, especially in the boreal region, will deteriorate the habitat quality of salmonid juveniles habituated to good freshwater quality. A reduced freshwater habitat connectivity can decrease the growth of juveniles and may have deleterious impacts on later marine life stages [104]. Flow regulation typically creates flow and temperature conditions for fish species that prefer warm- and slow-water habitats and can thus favor invasive species. The physically challenging migratory conditions caused by flow regulation combined with large diurnal temperature fluctuations can restrict the migration of salmonids by limiting their ability to recover from fatiguing exercise [62]. A rapid temperature rise also has a negative effect on the osmoregulatory performance of Atlantic salmon smolts [105].

## 4. Smolting

In salmonids, genetic diversity combined with developmental flexibility leads to numerous pathways to residency, migration, or maturation [106], and especially among Pacific salmon, there are also other phenotypes than smolts out-migrating rivers [107,108]. Anadromous salmonid juveniles transform from parr to smolts to prepare for downstream migration and entry into seawater. Physiological and behavioral changes take place in the spring when juvenile salmonids undergo smolting. Smolting and smolt migration are considered critical life-history stages essential for survival [5]. While still in fresh water, fish undergo a preparatory smolting process involving morphological changes as they become silvery and streamlined [2] (Figure 1). Behavioral changes include decreasing rheotactic and optomotor sensitivities and fish's station-holding abilities [9]. The photoperiod and temperature regulate physiological changes via their impact on the neuroendocrine system [2]. Thus, because the photoperiod remains the same at the same date and site each year, the temperature will be critical in determining responses to future climate change. Within the same river system, the distance to the sea does not seem to play a role; populations are closer or further from the sea smolts at the same time [109,110]. Waterflow and its variability as another major environmental factor can act more as a timer to initiate migration [2,7], for example.

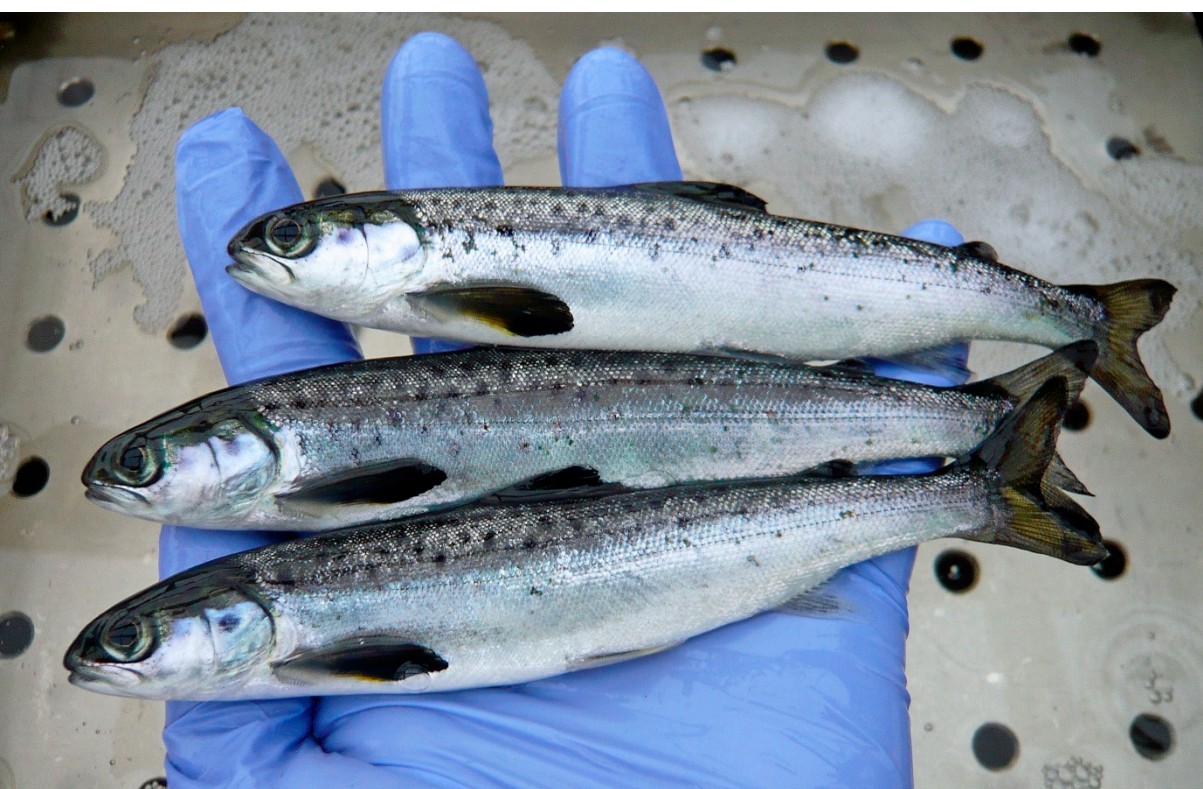

**Figure 1.** When smolting Atlantic salmon smolts become silvery and body-streamlined (Photo: River Tornionjoki (Finland) Atlantic salmon smolt, Ville Vähä).

Smolting varies depending on several factors like temperature, latitude, age and size, growth rate, and a combination of these factors. Hence, climate change with rising water temperatures obviously has an impact on the smolting process. Size and growth potentially affect the timing of migration and the survival of smolts [26,111–114]. Smolt age depends on growth rate. For instance, fast-growing parr smolts are younger and smaller than slow-growing parrs [115]. Warmer river temperatures increase the growth of parr and the share of fish smolting at an earlier age [74]. Temperature naturally correlates with latitude and is a strong predictor of migration timing in Atlantic salmon [116].

The migration decisions to smolt or not are decided between internal and external factors [114,117]. The important internal factors are the growth rate and the energetic status of individual fish [5]. Differences in the smolting rate between naturally anadromous and more resident populations have an inherited component [117]. A high growth rate in the late summer and the early fall of the year before migration can predict smolting [5]. However, high growth, especially during the winter, may induce the maturation of the parr [118].

Growth and energetics do not solely depend on temperature but on other factors like food availability. For example, it appears that smolting may be switched off via poor nutritional conditions preceding smolting [117,119]. Better growth conditions caused by an increase in the river temperature can increase the proportion of sexually mature male parr, which have a lower probability of migrating [120]. How climate change will affect individual growth rates and energetics in salmonid populations will depend intimately on how it affects the ecological status of rivers, particularly food availability.

Climate change may strengthen or weaken the effects of anthropogenic activities on water quality important for smolting salmonids. Pollutants, acidity, and sedimentation can adversely affect smolt development, which can have negative consequences on their readiness for life at sea [10,23]. Especially in northern temperate coastal regions, which will receive higher winter rainfall, phosphorus loading from land to streams is expected to increase, whereas a decline in warm temperate and arid climates is expected [121]. In the northern region, increasing precipitation will increase nutrient leaching, especially from areas affected by human alteration: agriculture, forestry, and other land use [122]. For example, acid leaks from the catchment are expected to increase. Increased acidity will have a major impact on the fish community, especially on acid-sensitive salmonids [123]. Even a short moderate exposure to acidity may require more than two weeks for the recovery of Atlantic salmon smolts [124]. Freshwater ecosystems are sensitive to anthropogenic flow regime alteration, which may cause temperature fluctuations. Close to its southernmost distribution, warming with low flows threaten coho salmon in California, and environmental flow protection is needed to support Pacific salmon in a changing climate [125]. Rapid temperature shifts have a negative impact on the hypo-osmoregulatory capacities of Atlantic salmon smolts [126]. There is an interaction of salinity and elevated temperature in the osmoregulatory performance of salmon smolt, and rapid temperature fluctuations above the threshold temperature (20 °C) have been found to cause iono-regulatory failure.

## 5. Smolt Migrations

Smolts start their downstream migration during a "period of readiness", a smolt window when they are physiologically prepared to meet the conditions in their marine feeding area [2,127]. In Atlantic salmon smolt, migration typically takes place during the spring and early summer at a length of 12–25 cm [128]. Temperature and flow are environmental cues for smolt migration. Migration times differ between years and rivers; the temperature can be a good predictor of the timing [129–131]. Warmer temperatures result in earlier migrations [130]. Typically, a correlation between the onset of the smolt run and the water temperature has been found [132]. Temperature experience, an accumulated temperature, or a combination of a temperature increase and temperature level in the river during the spring are the cues to initiate migration rather than any threshold temperature [130,133,134]. The initiation of smolt migration was positively associated with freshwater temperatures of up to about 10 °C and leveling off at higher values [18]. Another major environmental clue that plays an important role in initiating smolt migration is river flow. During the smolt window, increased waterflow initiates smolt migration [128,132,135,136], but high flows have also been found to have an opposite influence by depressing migration [130,132]. Depending on the conditions, the relative influence of water temperature and flow in initiating migration can differ across years [137]. Other environmental cues, like the photoperiod, have been found to control the initiation of downstream migration [138], but temperature and flow are the key environmental factors to be considered in response to climate change.

When ready, smolts lose their willingness to maintain station in a flow and start migrating downstream with the aid of the current. The speed of the current influences the downstream travel time, but smolts actively swim, typically following the mainstream in the surface water layer [7,120,139]. Smolts predominantly migrate at night, but this may change later in the migration period [2,5,140]. Smolts migrate downstream in schools of varying sizes. Relatively little is known about the formation of these groups. Some results indicate solitary movement from natal streams, followed by schooling further downstream [141]. A genetic component is involved as Atlantic salmon smolts migrate more in kin-structured groups than with unrelated individuals [142]. Some environmental factors like light and dark variations can influence schooling [143].

The timing of migrations has been adapted via evolution to avoid unfavorable conditions and arrive when environmental conditions are suitable for survival and growth [4]. Mismatched timing would lead to decreased fitness, depleted food sources, and/or increased predation. As described above, the environment has an effect on migration timing, but it is also influenced by inheritance [144,145]. The relative contribution of genetic differences remains uncertain [146]. Under climate-induced environmental changes, different migrating phenological traits may be important for the fitness of individuals [128]. It is obvious that Atlantic salmon migration timing is already responding to warming temperatures: The initiation of a smolt's seaward migration has occurred approximately 2.5 days earlier per decade throughout the basin of the North Atlantic [18]. Accordingly, the long time series analysis (1978–2008) of the timing of the smolt migration of Atlantic salmon in the River Bush, Northern Ireland, revealed that earlier downstream migration periods were evident across the time series [147]. Kastl et al. [125] found that an increase from 10.2 to 12.8 °C in mean seasonal water temperature accelerated the migration window by three weeks in coho salmon living near its southern distribution range in California, USA.

The earlier migration timing has given rise to growing concerns about smolts potentially missing the optimal environmental migration "window" [23]. Global warming also affects the receiving marine ecosystem by increasing surface seawater temperatures, and the results of this mismatch are difficult to predict. Climate change affects how and when species interact, potentially decoupling species interactions, combining others, and reconstructing predator–prey interactions [148]. Some of these mismatches may lead to increased predation on smolt production or cause starvation; some may have no effect. A better understanding of how these interactions work is crucial to predict vulnerability to the effects of climate change. Monitoring the timing and number of migrating smolts is important for revealing the effects of the changing climate on the smolt run. The quantification of migrating smolts to produce assessments of possible changes in natural reproduction, rates of survival, and patterns of migration, for example, by smolt trapping, is important for management (Figure 2).

The changed timing of smolt migration may lead to long-term changes to the migratory phenotypes of salmonids, e.g., [4]. A wide migration window with a diversity of phenotypes can act as a safeguard against uncertainty in resource availability, buffering the variability in predator pressure or thermal mismatch. The survival of phenotypes can depend on seasonally fluctuating conditions, such as thermal or hydrological circumstances affecting food availability, either directly or indirectly [149]. For example, Sturrock et al. [108] found that relative proportions of migrating phenotypes that contributed to the spawning population differed between the wet and dry years in chinook salmon. In California's chinook salmon, the late migrating phenotype dominated, but other strategies played an important role for many years [76]. Kennedy and Crozier [147] observed that the marine survival of one sea winter Atlantic salmon was strongly influenced by the run timing, and during the observation period, later emigrating cohorts demonstrated increased survival. In lake-migrating sockeye salmon entering Lake Washington, juveniles migrating later in the season encountered higher zooplankton abundance and warmer water, but the optimal date for lake entry ranged across years by up to a month [149]. These examples show that the success of migratory phenotypes varies with environmental conditions. The warm-

ing of waters may highlight the importance of rare phenotypes in responding to climate change [76]. The loss of phenotypic diversity can, therefore, have an impact on population persistence in a warming climate.

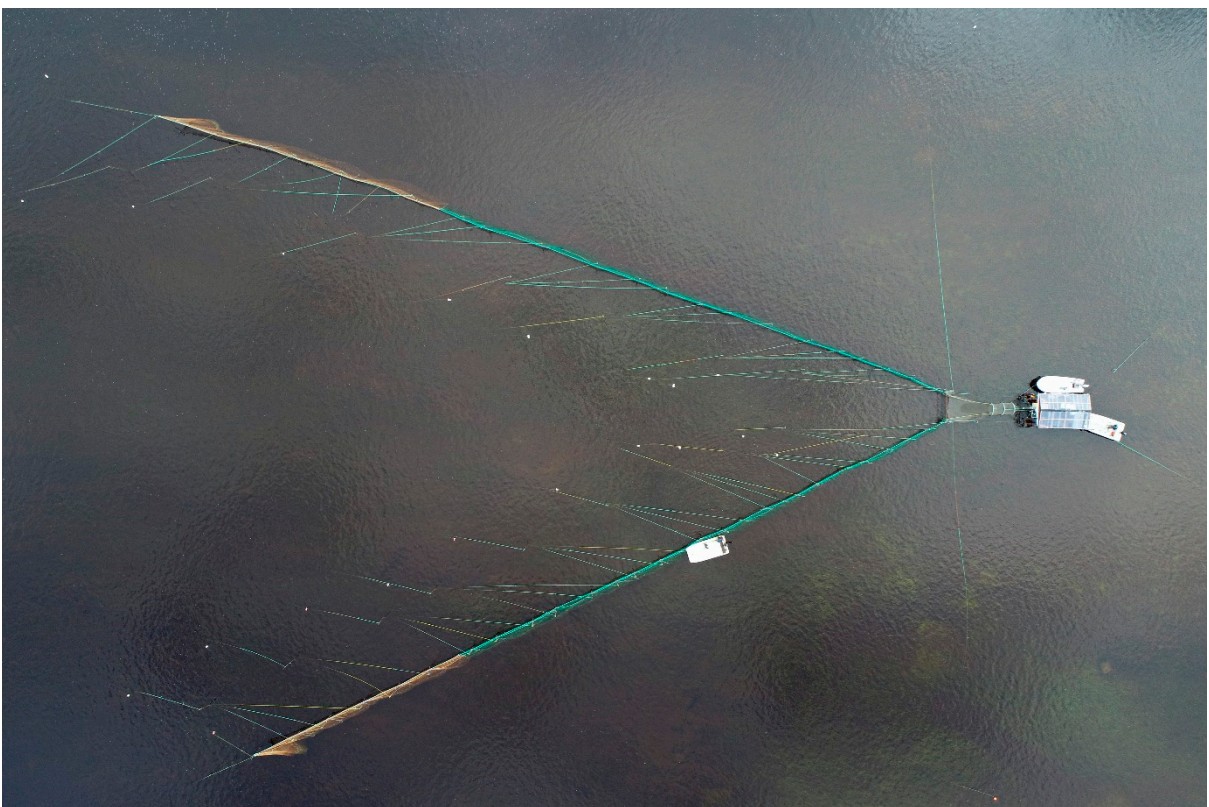

**Figure 2.** A smolt trap in the River Tornionjoki, Finland, to catch downstream migrating smolts and monitor their annual numbers and condition (Photo: Ville Vähä).

Anthropogenic use of fresh water, especially flow regulation with dam construction, has resulted in population declines and a loss of salmon life-history diversity [150,151]. Anthropogenic pressures, including climate change, affect the selection pressures in migratory salmonids, including on migrating smolt phenotypes. It would, therefore, be difficult to consider potential evolutionary responses to climate change without considering other human effects. For example, migration route selection at a hydropower plant intake has been found to be consistent with phenotypes, and those traits selecting turbines could potentially be eliminated from the population due to high mortality [152]. A long spill time is needed to protect the earliest and latest migrating phenotypes [131]. Low flows are expected to become more frequent, especially in the southern distribution area of salmonids, and during low flows, even small weirs can cause significant delays in smolt migration impacts [153]. River regulation practices are affected by climate-induced changes in temperature and precipitation, depending on the region, and they may also change selection pressures, affecting salmonid populations.

Survival during migration and patterns of mortality have the potential to yield important insights into population bottlenecks [154]. Smolts are vulnerable to predators during their downstream migration in the surface layer. In the southern River Minho (Spain/Portugal) and in the River Endrick (Scotland), the mortality of Atlantic salmon smolts by avian and piscine predators was high, demonstrating that the number of smolts lost in the river is likely to constrain population abundance in these rivers [154,155]. High in-river mortalities during downstream migration have also been found in Pacific salmon [156]. Climate change may create conditions that allow the successful spread of predators, including invasive species [157–159]. For example, increasing predator populations of cormorants

(*Phalacrocorax carbo sinensis* (Staunton, GL 1796)) and an invasive terrestrial predator in Europe, American mink (*Neovison vison* (Schreber, 1777), can cause elevated predation pressure on smolts [160–162]. Further, more detailed studies quantifying the impact of invasive species and climate change on smolt migration are needed for future management considerations.

## 6. Early Post-Smolt Survival

Most mortality between the smolt and adult stages is generally considered to occur during the first year of life at sea when survival, maturation, and migration trajectories are being defined [40,46,163,164]. Salmon's first year at sea, known as the post-smolt year, is characterized by variable mortality rates [165]. Mortality has often been considered to be highest during the first few months at sea [166,167]. Young salmonids are sensitive to variable climatic factors and food availability [168–170]. Reduced marine survival is widely accepted as an important contributor to the observed salmon population declines in recent decades [24,40,171,172]. Ocean climate variability during the first spring months of juvenile salmon migration to the sea seems to be central to the survival of North American populations, whereas summer climate variation appears to be important to adult recruitment variation for European populations [165]. In the Baltic Sea, marine survival estimates of salmon post-smolt were negatively correlated with temperature [173]. The anticipated warming due to global climate change will impose thermal conditions on salmon populations outside the historical context and will challenge the ability of many populations to persist [165].

The timing of salmon smolts' seaward migration and the size of smolts must be balanced with the marine conditions for the successful fulfillment of the life cycle [18,23,174]. Smolts' seaward migration should coincide with optimal thermal conditions at sea to maximize survival [2,40,175], but climate change has advanced the timing of salmon smolt migration and created a mismatch with optimal conditions for post-smolt growth and survival [18,147,176,177]. In the Gulf of St. Lawrence in the northwest Atlantic, the survival of sea-entering small smolts was found inferior to that of large smolts [178]. Smolt size can also influence the subsequent growth rate of Atlantic salmon at sea, with larger smolts showing slower growth [179]. Observations on brown trout in the River Imsa, Norway, suggest that an increased water temperature will induce seaward migration in the early spring, when sea growth and survival are poor [170].

Warmer temperatures in the North Atlantic have modified oceanic conditions, reducing the growth and survival of salmon by decreasing marine feeding opportunities [40,46,180,181]. Spring plankton blooms and, therefore, the peak of higher trophic resources available for salmon may be advanced in the season and may occur in different places [182–184], potentially creating a mismatch between salmon smolt migration and available resources [172,185]. A climate-driven shift in the zooplankton community composition towards more temperature-tolerant species with limited nutritional content may be associated with the decreased marine survival and growth of salmon smolts [169].

On a local scale, controlling climate change drivers is impossible. Proactive measures against population decline are, therefore, recommended [186]. These measures can include sustainable land use in the catchment and maintaining a diverse mosaic of habitats for salmonids [76,186]. Catchment scale conservation, including flow and connectivity restoration, is an important management priority for maintaining and improving juvenile salmonid, and thus smolt, production. Conserving headwater stream habitats maintains and increases the variability in habitats and the life history of salmonids to mitigate the effects of climate change. The freshwater environment is especially vulnerable to climate change effects because it is already exposed to numerous anthropogenic pressures, and water temperature and flow are highly climate dependent [187,188]. To secure the adaptive variability of a smolt, a safe downstream passage should be ensured at hydro dams, either with physical structures or sufficiently long spill time windows. Improved flow management is needed under climate change to avoid a further loss of phenotypic diversity in

salmonids. It is important to integrate management throughout the life cycle, including both sea- and freshwater phases, to secure a positive outcome for the salmonid populations (Figure 3).

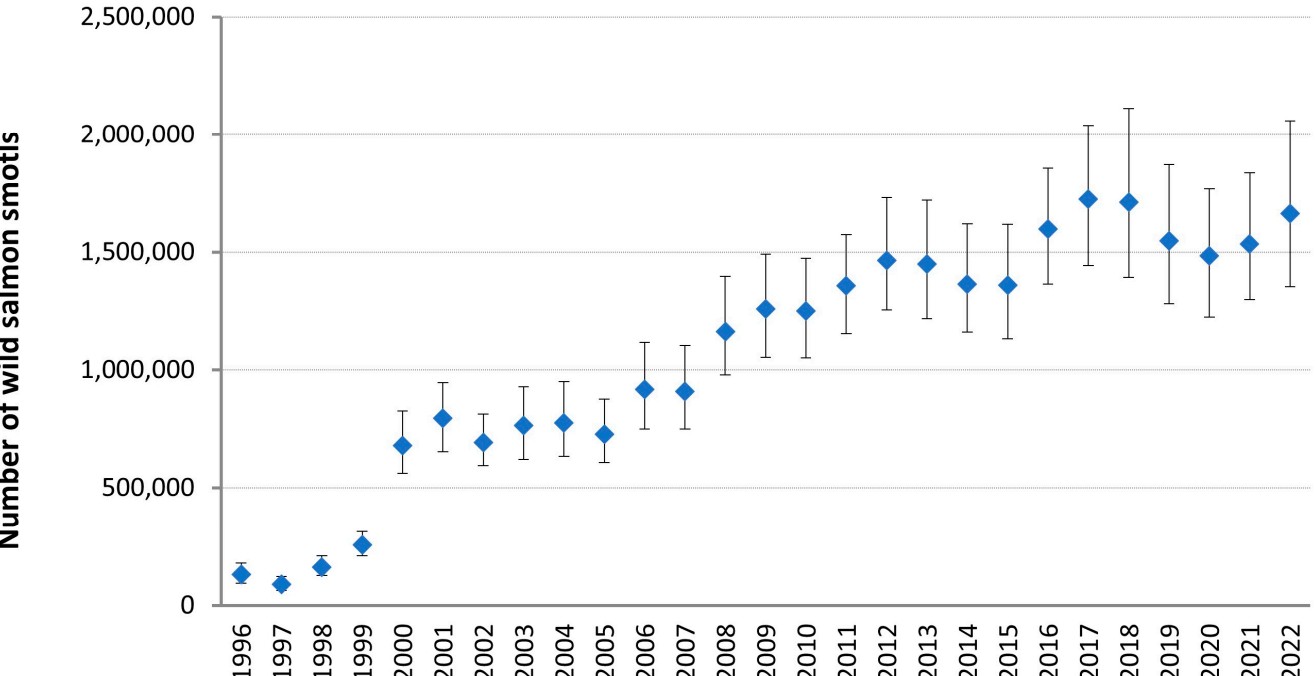

**Figure 3.** Toward a better future: After successful management actions in the Baltic Sea and in the river itself, the annual number of smolts migrating from the River Tornionjoki has increased substantially [189].

## 7. Conclusions

Atlantic salmon are already experiencing and responding to climate-change-induced warmer water temperatures at different scales. On a large scale, there are signs of salmon expanding their range northward, as expected due to a shift in the suitable thermal habitat, while southern populations are struggling more due to high temperatures and periodic droughts. While an increased temperature can have positive effects on the production of northern populations, increasing growth leading to earlier smolting also results in concerns about increased water quality problems via water brownification and eutrophication, particularly from human-impacted land areas with the effects of increased rain, especially during the winter. The risk of local extinction of salmon populations has increased, especially at the southern edge of salmon distribution. Low flow events, especially those typically associated with high water temperatures, are population bottlenecks. Mitigating the effects of climate change on a local scale to increase smolt production includes flow management and precautionary efforts to maintain and improve the ecological status of rivers. These measures are land-use planning and restoration on a catchment scale to diminish loading from the catchment and promote the conservation and restoration of instream habitats. Dense forests along riverbanks can decrease the water temperature.

Smolt characteristics depend in many ways on the factors acting in fresh water, and these characteristics affect post-smolt survival in the feeding area. Thermal heterogeneity in the river plays a significant role in survival and growth, and we should have better knowledge of the magnitude and location of cold-water refuges in streams. Mapping these areas with modern technology would help in conservation work. Maintaining a diverse mosaic of habitats and connectivity via conservation and restoration is crucial for mitigating climate change effects in rivers.

In response to increasing temperatures, an earlier migration timing of smolts is evident throughout the range of salmonids. This changes how and when species interact. It also

restructures predator−prey interactions. To readjust to the changed, and still changing, conditions, it is important to maintain the widest possible smolt window to allow all existing phenotypes, whether early or late migrants, to prevail. Under climate change, different migrating phenological traits may be especially important to the future fitness of the species. This is especially important in regulated rivers, where the anthropogenic alteration of waterflow creates not only increased mortality but also artificial selection pressure on migrating smolts. For example, this would mean longer spill water times or keeping the downstream routes open throughout the migration period.

Predation creates a substantial impact on migrating smolts and, thus, on the entire population. Climate change enhances the spread of invasive species, including invasive predators, which can increase the total predation pressure on smolts. This emphasizes better control of invasive species, the prevention of their dispersal, and better control of their populations.

Finally, we agree with the previous literature, stating that collaboration and research among scientists and managers across life cycle stages and ecosystems are urgently needed to address the research gaps [27] and that the basic strategy to protect salmonids against the effects of climate change should be to ensure that the maximum number of wild smolts in the best condition leave rivers, e.g., [23,24].

**Author Contributions:** Conceptualization, T.V., T.S. and A.H.; methodology, T.V., T.S. and A.H.; resources, T.V., T.S. and A.H.; data curation, T.V., T.S. and A.H.; writing—original draft preparation, T.V., T.S. and A.H; writing—review and editing, T.V., T.S. and A.H.; visualization, T.V., T.S. and A.H.; supervision, T.V.; project administration, T.V.; funding acquisition, T.V. All authors have read and agreed to the published version of the manuscript.

**Funding:** This research was funded by the Natural Resources Institute Finland.

**Data Availability Statement:** No new data were created or analyzed in this study. Data sharing is not applicable to this article.

**Conflicts of Interest:** The authors declare no conflict of interest.

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
