# Peer review of "Potential Impact of Climate Change on Salmonid Smolt Ecology"

_fishes, doi:10.3390/fishes8070382_

Round 1
Reviewer 1 Report
Congratulations for your work results.
For the moment not all the NOToriginal texts/info/data have the proper source citation in the paper.
The introduction can be more synthetic as far as some of the information are repeated extensively in the paper other main chapters.
More accurate biogeographic situation and potential trends can be revealed based on this study.
Different salmonids taxa are somehow mixed in the paper. The authors can find a solution to signal for the reader more obvious when a group/taxa or another is approached, in a logical way along the paper. So different taxa related info can be easily found by a reader in hurry without to read all the paper. Can help in the future citations of the paper.
Putting sometimes in the discussions and comments the salmonids in a more wide fish fauna context (without of course to go in details) can integrate the results in such a way in which the paper to be more relevant as approach and results.
All the best
Reviewer 2 Report
The authors reviewed a very hot-topic, providing a huge amount of literature. In general, however, as demonstrated by the available literature, this topic is not so new and the effort made by the authors in collecting all this information should be valorised.
First of all, I suggest to add a paragraph before the Conclusions reporting the proposed mitigation strategies. Several sentences in the manuscript could be moved in this specific section, I highlighted some of them in the text.
Another general consideration: I suggest to use the term "population" instead of "stock" if not specifically referred to populations exploited by fisheries.
Moreover, I added specific comments in the pdf concerning the need of figures and tables to make the manuscript more readable and clear. In particular, a table organized as follows: 1) developmental stage affected; 2) changing environmental condition/parameter; 3) effect (physiological, behavioural, morphological); 4) literature.
When paragraphs are particularly complex, I suggest the use of sub-headings.
Please, don't use the term "emigration".
All other comments are in the attached pdf.

English language is generally fine.
Reviewer 3 Report
The MS reports an exhaustive review of the impact of climate change on salmonid smolt ecology. Being a review article, the MS reports all the literature findings and represents a useful reference for future studies. The MS is well-structured and scientifically sound. There are a few minor revisions to carry on before considering the MS for publication. They are all indicated in the pdf attached but to summarize, I suggest changing the title by adding the word ''potential'' before impact. This is because it is not always possible to understand whether the impacts are only due to climate change or by the synergy of different factors/threats.
Another issue is the name of the Pacific salmon that is given as (Onchorhynchus sp.). The authors should add to the MS the reason for not using the full name of the species (if any).
Another point is the distribution of Salmo trutta which is reported in the MS as ''native to Europe and Asia where anadromous populations are found from Portugal to the White Sea''. Some sentences based on the most updated literature should be added to revise the new distribution of the species in line with the report that some populations of Salmo trutta in Europe belong to other species than S. trutta.
Last, reason for the selection of only a few salmonids species should be given in details in the abstract and in the aim of the study.

Round 2
Reviewer 2 Report
Authors addressed the issues that I raised. I don't agree with them, I guess that a table will really help summarizing the results obtained from other studies, this is not rare in the scientific literature and in particular in reviews. It's for sure a big effort, but not a duplication.
I still suggest to remove figure 2, or to explain it better, because it is a little bit out of topic in a review.
English language is fine.
Author Response
Response to reviewers:
I still suggest to remove figure 2, or to explain it better, because it is a little bit out of topic in a review.
We now explain the Figure 2 better in the text to link it closely to the topic.
Authors addressed the issues that I raised. I don't agree with them, I guess that a table will really help summarizing the results obtained from other studies, this is not rare in the scientific literature and in particular in reviews. It's for sure a big effort, but not a duplication.
We agree that this would be helpful, and that it would be a huge effort. Maybe this approach could be a starting point for a new review article in the journal requiring some months of review work.
Best Regards,
Teppo Vehanen & authors